# Spatial heterogeneity and infection patterns on epidemic transmission disclosed by a combined contact-dependent dynamics and compartmental model

**Youyuan Zhu**[1,2,3], **Ruizhe Shen**[1,2,3], **Hao Dong**[1,3,4]*, **Wei Wang**[2,3,5]*

**1** Kuang Yaming Honors School, Nanjing University, Nanjing, China, **2** Collaborative Innovation Center of Advanced Microstructures, National Laboratory of Solid State Microstructure, Nanjing, China, **3** Department of Physics, Nanjing University, Nanjing, China, **4** State Key Laboratory of Analytical Chemistry for Life Science, Nanjing University, Nanjing, China, **5** Institute for Brain Sciences, Nanjing University, Nanjing, China

* donghao@nju.edu.cn (HD); wangwei@nju.edu.cn (WW)

**Data Availability Statement:** All relevant data are accessible through this DOI:(10.5061/dryad. qrfj6q5mp).

**Funding:** This work was supported by the National Natural Science Foundation of China (Grant Nos.

## Abstract

Epidemics, such as COVID-19, have caused significant harm to human society worldwide. A better understanding of epidemic transmission dynamics can contribute to more efficient prevention and control measures. Compartmental models, which assume homogeneous mixing of the population, have been widely used in the study of epidemic transmission dynamics, while agent-based models rely on a network definition for individuals. In this study, we developed a real-scale contact-dependent dynamic (CDD) model and combined it with the traditional susceptible-exposed-infectious-recovered (SEIR) compartment model. By considering individual random movement and disease spread, our simulations using the CDD-SEIR model reveal that the distribution of agent types in the community exhibits spatial heterogeneity. The estimated basic reproduction number $R_0$ depends on group mobility, increasing logarithmically in strongly heterogeneous cases and saturating in weakly hetero-geneous conditions. Notably, $R_0$ is approximately independent of virus virulence when group mobility is low. We also show that transmission through small amounts of long-term contact is possible due to short-term contact patterns. The dependence of $R_0$ on environ-ment and individual movement patterns implies that reduced contact time and vaccination policies can significantly reduce the virus transmission capacity in situations where the virus is highly transmissible (i.e., $R_0$ is relatively large). This work provides new insights into how individual movement patterns affect virus spreading and how to protect people more efficiently.

## Introduction

The outbreak of COVID-19 brought a profound disaster to human civilization worldwide, which has infected more than 600 million people and killed more than 6 million people by October 2022, and the virus is still in constant mutation [1, 2]. From the original strain to the

12090052, 21833002), the National Key R&D Program of China (2021ZD201302), and the "Fundamental Research Funds for the Central Universities" (021514380018). Parts of the calculations were performed using computational resources on an IBM Blade cluster system from the High-Performance Computing Center (HPCC) of Nanjing University. The funders had no role in study design, data collection and analysis, decision to publish, or preparation of the manuscript.

**Competing interests:** The authors have declared that no competing interests exist.

Alpha, Delta, and Omicron variants, while the virulence of the virus decreased, the infectivity gradually increased. As an important indicator to characterize infectious diseases, the basic reproduction number ($R_0$) has increased from 2.79 in the original strain to 9.5 in the Omicron variant [3–5], bringing new challenges to epidemic prevention and control.

In the theoretical models of studying the transmission of infectious disease, homogeneous mixing [6] is a classic assumption of the traditional compartmental model [7]. This approximation simplifies the model and thus can help to obtain epidemic transmission trends more efficiently. However, in real situations, heterogeneity, including differences between individuals [8–10] and differences in spatial distribution [11–14], is also an important factor in epidemic transmission that should be considered [8–14].

In recent years, agent-based models (ABMs) that focus on individual behavior (even considering the corresponding movement characteristics) have received increasing attention [15, 16]. This model examines the overall behavior and trends by modeling many individuals (called agents). In ABM, individuals are made autonomous and adaptive by defining them to choose behaviors based on their own decision rules (often derived from limited information in their immediate surroundings), and the decision rules could vary from agent to agent. Agents are social and interdependent. If an agent performs a certain action, other agents may be influenced and respond in some way. A central concept of ABM is that the overall characteristics of a group are not simply the sum of all individual characteristics in the group but are the result of interactions guided by individual characteristics. Interactions among individuals often result in a society with a different overall identity from that of the individuals. In the case of epidemic transmission, the spatial movement of individuals and the influence of spatial heterogeneity on the spread of an epidemic are important factors that cannot be ignored. Adopting the idea of ABM to study may be an appropriate idea [12, 14, 17–23].

In some ABMs, the location and interdependence between individuals are determined by a predefined social network (e.g., building a small-world network to simulate the social relationships of agents), and the spread of epidemics is demonstrated through the evolution of the network [24, 25]. Complex network models can better introduce heterogeneity among individuals but lack the spatial movement of individuals [19]. In addition, there are individual-based motion models that rely not on defined social networks but only on individual motion [26, 27]. These models suggest that characteristic quantities of motion, such as the velocity of individuals, can play an important role in the spread of the epidemic [19, 20]. However, these models often simplify factors related to epidemic transmission because of computational efficiency [12, 14, 17–22]. For example, the motion and the motion features of an individual at a real time-space scale are not fully considered.

In this work, we designed a contact-dependent dynamics (CDD) model based on the characteristics of epidemic transmission to mimic the motion of the population within a community. Our model consisted of alternating individual motions and contact between individuals (thus changing the direction of motion) in an ensemble of individuals. We combined this CDD model (to describe population movement) with the commonly used epidemic transmission model, the susceptible-exposed-infectious-recovered (SEIR) model (to describe infection between individuals), to study the characteristics of epidemic transmission by studying the movement of individuals at real time and space scales and epidemic transmission. We found that the CDD model has a larger proportion of short-term contacts as the velocity of agents decreases. To characterize the epidemic transmission capacity, we calculated the basic reproduction number $R_0$ of the model. The calculated $R_0$ with the CDD-SEIR model is nearly independent of the virulence of the virus and much less than that calculated by mean-field equations [19], which is attributed to the short exposure time $\omega$ of the CDD model when the velocity of all individuals in the population is less than a threshold value. This corroborates the

effectiveness of prevention and control measures in the presence of highly infectious strains. Moreover, we found that $R_0$ shows two stages of variation with the spatial heterogeneity of the system: that is, in the strong heterogeneity range, $R_0$ is proportional to the logarithm of velocity, which is different from mean-field theory, and in the weak heterogeneity range, $R_0$ reaches saturation. The above results provide us with a new understanding of the relationship between epidemic transmission and individual movement.

This paper is organized as follows: we first propose a modified SEIR model to incorporate individual mobility. Then, we introduce the CDD model designed to characterize the stochastic motion of individuals within a community. Next, we describe the CDD-SEIR model, combining real-scale stochastic individual motion with epidemic transmission. To characterize the agent dynamics, the exposure time $\omega$ and mean free time $\tau$ will be discussed. Meanwhile, we show the characteristics of epidemic spreading described by this CDD-SEIR model, as well as the calculated basic reproduction number $R_0$ and the spatial heterogeneity $\zeta$, obtaining the effect of motion parameters (e.g., velocity) on these properties. We will then discuss the bipartite variation of $R_0$ with the strength of spatial heterogeneity. Notably, $R_0$ is nearly independent of the virulence of the virus when all individuals move at less than a threshold value and spatial heterogeneity is sufficiently strong. $R_0$ is much less than that calculated by mean-field equations due to the short exposure time caused by the CDD model. Therefore, an in-depth understanding of heterogeneity is important for an accurate understanding of epidemic transmission, e.g., prevention and control measures are still effective when the virulent strain is highly infectious due to the presence of spatial heterogeneity. The short-term contact model can greatly suppress the epidemic when combined with control measures.

## Methodology

### The SEIR model

The SEIR model is one of the compartmental models that is commonly used to predict epidemic transmission [28]. Here, S, E, I and R stand for susceptible, exposed, infectious, and recovered, respectively, which are the four categories of people in epidemic transmission. The sum of these four is the population within the community. In the SEIR model, susceptible people who become infected come through exposed, infectious, and recovered compartments successively, as shown in **Fig 1A**. The latent period was considered when individuals had been infected but were not yet infectious themselves.

COVID-19 has been reported to be transmitted mainly by proximity contact [29]. To introduce the mobility of individuals into the study of epidemic transmission, we proposed a modified SEIR model. According to the definition of the SEIR model, the agents here fall into four categories: susceptible, $P_S$, healthy individuals who may be infected; exposed, $P_E$, individuals who have been infected but do not cause infection after an incubation period; infectious, $P_I$, individuals who have been infected and can transmit the virus to $P_S$; and recovered, $P_R$, individuals who have been cured or immunized and are thus removed from the transmission process.

In the modified SEIR model (**Fig 1B**), the $P_S$ has a probability $p_{SE}$ to convert to $P_E$ when the distance $r$ between the susceptible $P_S$ and the infected $P_I$ is less than the airborne transmission distance $d$. The latent period for COVID-19 was estimated to be approximately 7 days in the literature [2, 30]. Accordingly, we set the latent period of $P_E$ between 4 and 10 days. Specifically, within 0–4 days, $P_E$ only acts as the virus carrier and does not have the ability to infect others; within 4–10 days, $P_E$ can transform with the probability $p_{EI}$ (Eq 1, where $T$ is in the unit day) to $P_I$, which can transmit the virus; otherwise, it remains as $P_E$. After 10 days, $P_E$ that still does not convert to $P_I$ is classified as $P_R$. Meanwhile, $P_I$ can be transformed into $P_R$ with

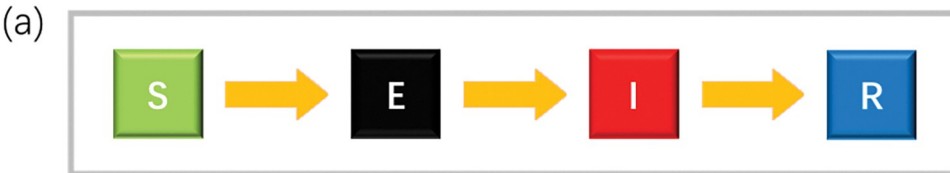

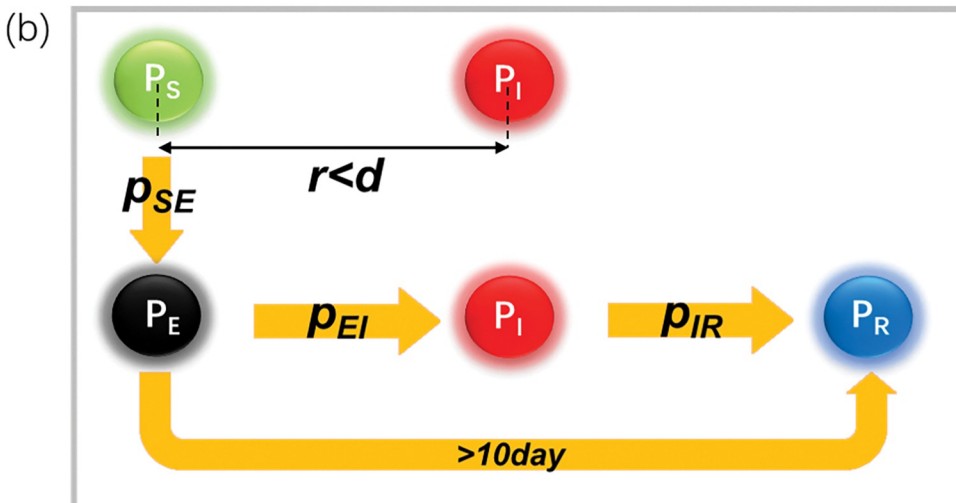

**Fig 1. The classic SEIR model and the modified SEIR model.** (a) The classic SEIR model. Green, black, red, and blue squares represent susceptible, exposed, infectious, and recovered compartments, respectively. Susceptible individuals in the system successively go through these four stages. Transitions between adjacent stages occur according to certain probabilities. (b) The modified SEIR model. When the distance between the susceptible $P_S$ and $P_I$ is within the airborne transmission distance $d$, $P_S$ can be infected by $P_I$ and transformed into $P_E$ at the probability $p_{SE}$. The latent period is within 4–10 days. During this period, $P_E$ converts with probability $p_{EI}$ to infectious $P_I$ that can transmit the virus. Any $P_E$ that remained unconverted to a $P_I$ after 10 days is classified as a recover $P_R$. Meanwhile, $P_I$ could also be cured with probability $p_{IR}$ and transformed into $P_R$.

the probability $p_{IR}$.

$$p_{EI} = \begin{cases} 0 & T < 4 \\ p_{EI,Day7} * \dfrac{2}{1 + e^{3(T-7)}} & 4 \leq T < 10 \end{cases} \tag{1}$$

The conversion probability from the exposed agent ($P_E$) to the infectious agent ($P_I$) $p_{EI}$ is reduced with the increase in Day $T$ from latency, as shown in Eq 1. $p_{EI,Day7}$ is the conversion probability of the 7th day of the latent period, and its value is shown in **Table 1**.

## The CDD model

We introduce the CDD model to characterize the motion of agents. In CDD, the agents are described by points. Each agent $i$ is assigned a velocity $v_i$ following a normal distribution with a mean of $v$, a standard deviation of $\frac{1}{3}v$, and a direction $\theta_i$ randomly generated from a uniform distribution. The motion of the agents consists of alternating individual motions and contact between individuals (thus changing the motion) in an ensemble of individuals.

When there are no other agents within the distance $r_c$ around agent $i$, the latter adopts uniform linear motion, and the evolution of its position is determined only by its velocity; otherwise, the direction of motion and the velocity of $i$ will be regenerated, as shown in **Fig 2A**,

**Table 1. The different mobility and propagation parameters tested in this model.** We selected 9 different velocities $v$ and 5 different values for the remaining parameters, including the initial proportion of $P_I$ $n_{I,0}$, the airborne transmission distance $d$, the infection probability $p_{SE}$, the conversion probability of Day 7 $p_{EI,Day7}$ from $P_E$ to $P_I$, and the recovery probability $p_{IR}$. Those marked with * are the default values. In each change of one parameter, the default values were used for the other parameters. For each set of parameter combinations, we calculated 5 random motion trajectories of agents at 40 days length separately using the CDD model and postprocessed each trajectory using the SEIR model to obtain 5 independent modes of transmission. The final reported results were extracted from the average of 5*5 = 25 trajectories.

| Parameters | Values |
|---|---|
| $v$ ($10^{-4}$m/s) | 1, 2, 5, 10, 20, 50, 100, 200, 500 |
| $n_{I,0}$ | 1%*, 3%, 5%, 7%, 9% |
| $d$ (m) | 1, 2*, 3, 4, 5 |
| $p_{SE}$ ($10^{-4}$) | 0.5, 1.0, 1.5, 2.0*, 2.5 |
| $p_{EI,Day7}$ ($10^{-6}$) | 1, 2*, 3, 4, 5 |
| $p_{IR}$ ($10^{-6}$) | 1, 2*, 3, 4, 5 |

when there are other agents within the distance $r_c$ around $i$ (Eq 2). In this case, the velocity is randomly generated by the normal distribution, and the direction of motion is randomly reassigned by the normal distribution within the range of ±45 degrees from the original motion direction. To represent the real-scale motion, the units for length and time used in this work were in meters and seconds, respectively. We set $r_c$ to 2 m and used the time interval $\Delta t$ of 1 s.

$$v_i(t + \Delta t), \theta_i(t + \Delta t) = \begin{cases} v_i(t), \theta_i(t) & \text{no contact} \\ v_{rnd}, \theta_i(t) + \theta_{rnd} & \text{with contact} \end{cases} \quad (2)$$

where $v_i(t)$ and $\theta_i(t)$ denote the velocity and direction of particle $i$ at time $t$, respectively. $v_{rnd}$ is the reassigned velocity that satisfies a normal distribution with a mean of $v$ and a standard deviation of $\frac{1}{3}v$. $\theta_{rnd}$ is the change in the reassigned direction (in the range of -45˚ to 45˚), which satisfies a normal distribution with a mean of 0˚ and a standard deviation of 15˚. The distributions of $v_{rnd}$ and $\theta_{rnd}$ are shown in Fig 2B.

Notably, this CDD model has some similarities to the stochastic rotation dynamics (SRD) model [31, 32], which has been widely used to characterize fluid flow. The main difference between the two models is that in the CDD model, the individuals change direction and velocity of motion when the distance between two individuals is less than a certain range, rather than waiting until they are in full contact and collide. Such a model setup provides the possibility to study the ability of the virus itself to spread in space for the spread of the epidemic and for social distancing interventions.

## The CDD-SEIR model

We combined the CDD model (to describe population movement) with the modified SEIR model (to describe infection between individuals) to simulate epidemic transmission at real time and space scales.

As shown in Fig 3A, a community represented by a periodic square box with side length $L$ has $N$ individuals. The population is composed of susceptible $P_S$, exposed $P_E$, infectious $P_I$ and recovered $P_R$. The number of each type is $N_S$, $N_E$, $N_I$ and $N_R$, respectively, where $N = N_S + N_E + N_I + N_R$. In subsequent work, we take the early stage of the epidemic as the initial condition of the simulations, put only a small amount of $P_I$ in the system and set the rest of the particles as $P_S$. It should be noted that other factors, such as repeated infections, vaccination, etc., have potential influence on the transmission mechanism, and can be easily added to the current

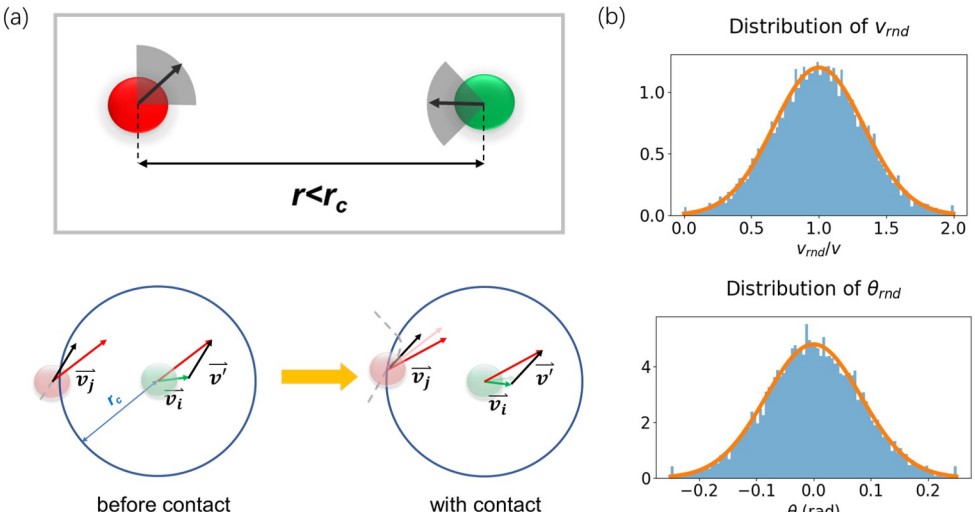

**Fig 2. Schematic diagram of the CDD model.** (a) Stochastic motion process characterized by the CDD model. The upper panel represents that the direction of motion of an agent varies within [-45˚, 45˚] of the original direction when there are other agents within $r_c$. The lower panel gives an example of the change in motion state of two individuals before and after contact: $i$ is the central agent, and the blue circle represents its contact range of radius $r_c$. Individuals $i$ and $j$ fall in contact when their distance is less than $r_c$, and the agents are in a uniform linear motion before contact. The red, green, and black solid vectors represent the velocities of $i$ and $j$ and the relative velocity $j$ relative to $i$, respectively. The gray dashed line represents the trajectory of $j$ relative to $i$. (b) The reassigned velocity $v_{rnd}$ that satisfies a normal distribution with a mean of $v$ and a standard deviation of $\frac{1}{3}v$. $\theta_{rnd}$ is the change in the reassigned direction (in the range of -45˚ to 45˚), which satisfies a normal distribution with a mean of 0˚ and a standard deviation of 15˚.

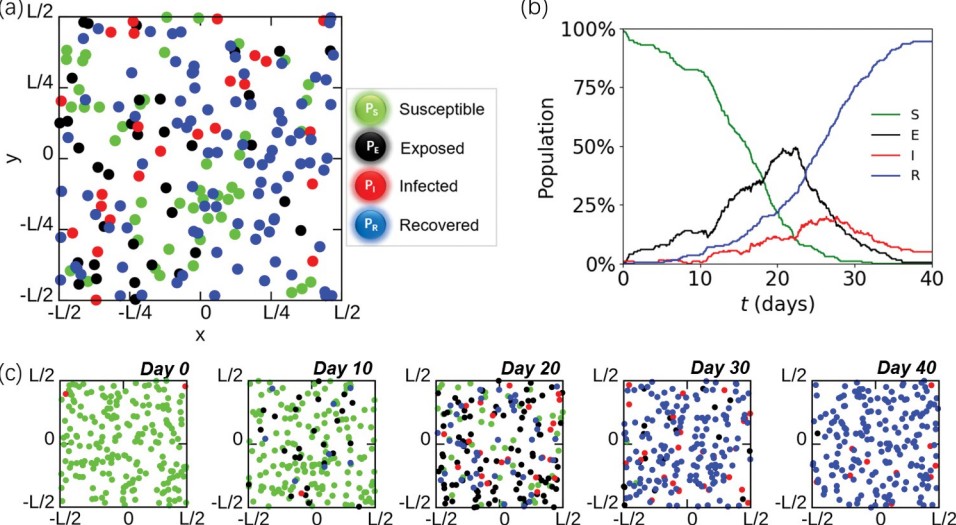

**Fig 3. System setup of the CDD-SEIR model.** (a) Schematic diagram of a representative system obtained from the simulations. The system contains 200 particles with a box length of $100\sqrt{2}$ m and density of $(100\ m^2)^{-1}$. $P_S$, $P_E$, $P_I$ and $P_R$ are represented by green, black, red and blue particles, respectively. (b) The profiles of the temporal population of $P_S$ (in green), $P_E$ (in black), $P_I$ (in red) and $P_R$ (in blue) during epidemic transmission. (c) Representative snapshots of the infection within a community at days 0, 10, 20, 30, and 40 are shown.

model (by adding types of agents). Meanwhile, factors such as death can also be implemented in the current model by removing particles from the system or eliminating their movement. The motion of agents is driven by the CDD model, and the spread of the epidemic is determined by the modified SEIR model. Therefore, we obtained the temporal evolution of the agents' populations (**Fig 3B**) and the dynamic progress of the infection spreading (**Fig 3C**). We can study the factors of epidemic spreading from the perspective of both spatial agent dynamics and epidemic dynamics.

Comparing the results obtained from the present CDD-SEIR model with the real epidemic data is a way to test the model's reliability. However, limited by the current model setup, such a direct comparison is not easy: first, we used 1 s as the integration step of the equation of motion to not miss any of the exposure events, and thus currently, the model can only handle a few hundred agents in a few tens of days within an affordable computational cost; second, for a system containing hundreds of particles, a single initial infected agent already corresponds to a very high initial infection rate (e.g., 1 in 200 agents corresponds to an initial infection rate of 2.5%), while the ratio in the real situation is often less than 1 in 10,000. Therefore, the present work lies mainly in providing a simple model with clear physical meaning to study the dynamics of epidemic transmission.

### Parameter selection of the CDD-SEIR model

Considering the computational efficiency, in the simulations, we used the following parameters for the CDD model: number of particles $N$ = 200, density of particles $\rho$ = (100 m$^2$)$^{-1}$, and contact distance $r_c$ = 2 m. These parameters remained intact in the simulations. Forty-day trajectories were accumulated separately under the different combinations of parameters.

Based on the case of COVID-19 propagation [2], we set the default values of some propagation parameters in the SEIR model to be close to the real situation. For example, we assumed the airborne transmission distance $d$ = 2 m. This is considered a safe social distance to prevent the spread of the virus [33]. In addition, we set $p_{EI,Day7}$ = $p_{IR}$ = 2*10$^{-6}$, which determines the duration of the incubation and disease periods, and assumed that $p_{SE}$ = 2*10$^{-4}$, which determines $R_0$ to be close to the real situation based on the reported data [30, 34, 35].

Meanwhile, we carefully examined the effects of the rest of the propagation parameters in the combined CDD-SEIR model, including the mobility $v$, the initial proportion of $P_I n_{I,0}$, the airborne transmission distance $d$, the infection probability $p_{SE}$, the conversion probability of Day 7 from the exposed agent ($P_E$) to the infectious agent ($P_I$) $p_{EI,Day7}$, and the recovery probability $p_{IR}$. By changing the value of the parameters at a time (keeping the other parameters at their default values), we systematically studied the effect of that parameter and the combination of different parameters on the spread of the epidemic. The values of the parameters are shown in **Table 1**.

## Results and discussion

### Agent dynamics and contact pattern

To investigate the effect of contact-dependent dynamics, we estimate the exposure time $\omega$ (the time that two particles stay in contact) and the mean free time $\tau$ (the time interval between two consecutive contacts) in the CDD model.

By counting the time that particles $i$ and $j$ remain in contact in trajectories, we can estimate the exposure time $\omega$ of different velocities in the CDD model. As shown in **Fig 4A**, the average of $\omega$ is inversely proportional to $v$, while the median decreases slightly as $v$ increases. This means that the agents do not intend to stay together for a long time as $v$ decreases. We classify contacts into short-term contact ($\omega < 0.5*r_c/v$), medium-term contact ($0.5*r_c/v \leq \omega < 4*r_c$

**Fig 4. Contact pattern of the CDD model.** (a) The mean (blue) and median (red) of the contact time $\omega$ influenced by the velocity $v$ of agents in the CDD model. (b) The percentages of $\omega$ in different time ranges for the velocity $v$ of agents in the CDD model. The blue line represents $\omega < 0.5*r_c/v$, the red line represents $0.5*r_c/v \leq \omega < 4*r_c/v$, and the green line represents $\omega \geq 4*r_c/v$. (c) The mean free time $\tau$ in the CDD model (solid line) and the CDD-NC model (dashed line) for different velocities.

$/v$) and long-term contact ($\omega \geq 4*r_c/v$) by the value of $\omega$ (**Fig 4B**). The results show that the CDD model has a larger proportion of short-term contacts as the velocity of agents decreases.

In addition, by dividing the total trajectory time by the average number of particle contacts obtained from the statistics of the CDD model, we can estimate the mean free time $\tau$. As a control, we also set up a contact-dependent dynamics with no contact (CDD-NC) model (**S1 and S2 Figs in S1 File**, please refer to the **S1 File** for more details), in which the velocities and directions of the individuals follow the same distribution in CDD and the motion does not change when two particles are in contact, as in the no contact case of Eq 2. We further estimated the mean free time $\tau$ (Eq 3) of the CDD-NC model.

$$\tau = \frac{1}{2r_c\overline{vr}\rho} \tag{3}$$

where $r_c$ is the contact distance, as defined before, $\overline{vr}$ is the average relative velocity and was set as $\overline{vr} = 1.35v$, and $\rho$ is the density of the system. With respect to $\tau$, we found that there was no difference between the two models (**Fig 4C**).

Therefore, the CDD model does not change the frequency of contact but only makes the contact pattern of individuals more inclined to short-term contact as the velocity of agents decreases. In the following, we found that this contact pattern caused strong spatial heterogeneity and determined the transmission pattern at low velocity.

## Epidemic dynamics and the basic reproduction number $R_0$

We examined the effect of five propagation parameters, including $n_{I,0}$, $d$, $p_{SE}$, $p_{EI,Day7}$ and $p_{IR}$ on epidemic evolution. For example, different $p_{SE}$ agents show different patterns of epidemic spread as shown in **Fig 5A and 5B**.

To reasonably characterize the epidemic transmission capacity obtained from the current CDD-SEIR model simulations, we defined the basic reproduction number $R_0$. Traditional models commonly use the basic reproduction number $R_0$ to indicate how contagious an infectious disease is. $R_0$ is the number of secondary infections an infectious $P_I$ can cause in a fully susceptible environment [36]. We calculated the basic reproduction number $R_0$ by the definition of $R_0$ in ordinary differential equations. Similar to SEIR model,

$$\frac{dN_s}{dt} = \beta\frac{N_I}{N}N_s \tag{4}$$

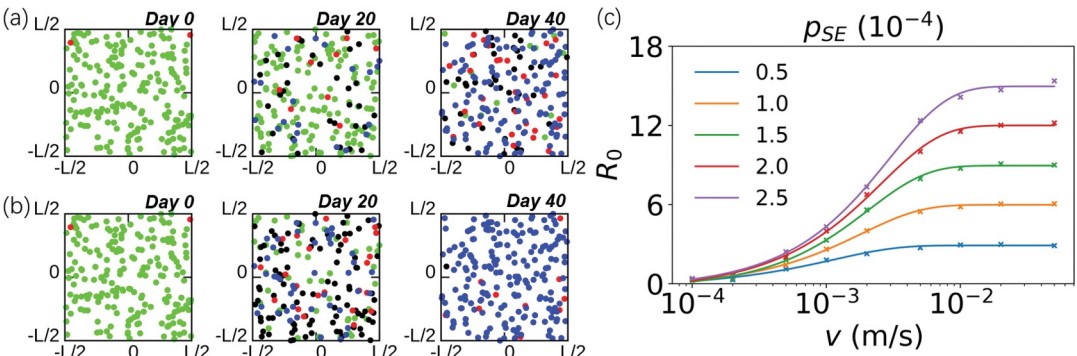

**Fig 5. Infection spreading in different $p_{SE}$ and the basic reproduction number $R_0$ influenced by $p_{SE}$ and the velocity $v$ of agents in the system.** The dynamic progress of the infection spreading when $p_{SE} = 1.0*10^{-4}$ (a) and $p_{SE} = 2.0*10^{-4}$ (b). Snapshots of Days 0, 20, and 40 are shown in order. (c) The influence of $p_{SE}$ and $v$ of agents on $R_0$ in the system. The points represent the calculated data, and the solid line represents the fitting curve to eliminate noise.

$$\frac{dN_R}{dt} = \gamma N_I \tag{5}$$

where $N_S$, $N_I$, $N_R$ and $N$ represent the susceptible, infectious, recovered and total populations of agents, and $\beta$ and $\gamma$ represent the transmission rate and recovery rate (which equals to $p_{IR}$ in our model), respectively. Then $R_0$ can be calculated by

$$R_0 = \frac{\beta}{\gamma} \tag{6}$$

We calculated the basic reproduction number $R_0$ in the CDD-SEIR model by

$$R_0 = \frac{\beta}{p_{IR}} = \frac{N}{p_{IR}dt}\frac{dN_S}{N_S N_I} \approx \frac{N}{p_{IR}\Delta t} < \frac{\Delta N_S}{N_S N_I} > \tag{7}$$

where $\Delta N_S$ represents the variation of $N_S$ in the integration time step $\Delta t$ (please refer to the **S1 File** for more details).

**Fig 5C** shows how $R_0$ is influenced by $p_{SE}$ and the velocity $v$ of agents in the system. In addition to $R_0$, we also counted the peak proportion of infections in the system, $n_{I,max}$, and the infection rate $p$ of the system when the epidemic had developed for 40 days to characterize spreading (**S4 Fig** in **S1 File**).

We examined the effect of five propagation parameters, including $n_{I,0}$, $d$, $p_{SE}$, $p_{EI,Day7}$, and $p_{IR}$, on $R_0$ when the agents in the system had a specific mobility. As shown in **Fig 6A**, except for the parameters $n_{I,0}$ and $p_{EI,Day7}$, $R_0$ shows a significant dependence on the other three parameters. Specifically, the ability of the virus to spread/infect is positively correlated with $d$, and $p_{SE}$ and negatively correlated with $p_{IR}$. This observation is consistent with our common perception: the larger the infection range ($d$), the more virulent the virus; the lower the recovery rate, the higher the severity of the outbreak. Meanwhile, it should be noted that these three parameters also showed different effects: in the speed range studied in the current work, $R_0$ shows a clear dependence on the infection range $d$, i.e., a larger $d$ led to a larger $R_0$; in contrast, $R_0$ is not significantly affected by different $p_{SE}$ and $p_{IR}$ when the system is under low mobility and only reflects notable differences when the agents are in high mobility.

We fit the curves for each of the two phases of $R_0$ linearly and define the critical velocity $v_c$ of $R_0$ at which $R_0$ reaches 90% of the saturated reproduction number (please refer to the

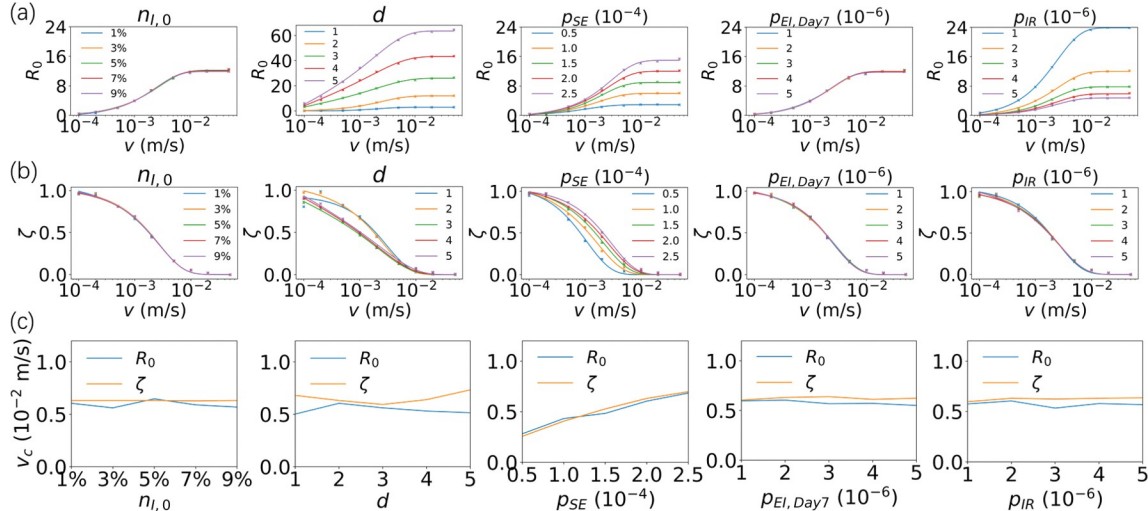

**Fig 6.** The basic reproduction number $R_0$ (a), the heterogeneity variable $\zeta$ (b) and the critical velocity $v_c$ of $R_0$ and $\zeta$ (c) influenced by five parameters ($n_{I,0}$, $d$, $p_{SE}$, $p_{EI,Day7}$, and $p_{IR}$) and the velocity $v$ of agents in the system. In each plot, the results corresponding to the five values used for the parameter that changed are represented by five different colored curves. The points represent the calculated data, and the solid line represents the fitting curve to eliminate noise. The horizontal coordinates in (a) and (b) are on a logarithmic scale.

**S1 File** for more details). The calculations show that when the mobility of the system is small (i.e., the individual's $v$ is less than $v_c$), $R_0$ is influenced almost exclusively by $d$, independent of other propagation parameters. When $v > v_c$, the system is close to the scenario obtained from the classical epidemic model assuming uniform mixing [6]. For example, in classical models, an infectious agent can come into contact with susceptible individuals at a probability proportional to $N_S$. Thus, the mobility of the infectious is not directly reflected in the process of virus transmission. However, when $v < v_c$, this is not the case. Epidemics transmitted by contact are closely related to the type of individuals who can be reached locally, the so-called spatial heterogeneity [12, 14].

## Heterogeneity variable $\zeta$ and spatial heterogeneity

**Fig 6A** clearly shows that the epidemic varies linearly with $v$ when $v < v_c$, indicating that heterogeneity significantly influences the spread of the epidemic; in contrast, the spread of the epidemic is no longer dependent on the variation of $v$ when $v > v_c$ [20]. Therefore, it is necessary to explore the factors influencing $v_c$. As shown in **Fig 6C**, we found that $v_c$ of $R_0$ is tightly associated with certain propagation parameters. Specifically, $v_c$ is positively correlated with $p_{SE}$, and nearly independent of $d$, initial $n_{I,0}$, $p_{EI,Day7}$, and $p_{IR}$. It should be noted that the independence between $v_c$ and the initial $n_{I,0}$ indicates that the epidemic is determined by its inherent properties. On the other hand, $p_{EI,\,Day7}$ and $p_{IR}$ do not affect the distribution of $P_S$ around $P_I$, which is associated with heterogeneity and thus $v_c$.

Apparently, $v_c$ indicates the boundary between heterogeneity and homogeneity. To further elucidate the above trend, we calculated the relative neighbor concentration $RNC_S$ of the system for characterizing spatial heterogeneity: $RNC_S$ was defined as the average of the ratio of the local concentration of $P_S$ within the infection range $d$ of a single $P_I$ in the system to the global concentration of $P_S$ in the whole system. The heterogeneity variable $\zeta$ was defined as 1-

$RNC_S$ (Eq 8).

$$\zeta = 1 - RNC_S = 1 - < \frac{N'_S/N'}{N_S/N} > \tag{8}$$

Here, $N'_S$ represents the total number of $P_S$ within radius $d$ of a $P_I$, $N'$ represents the total number of all individuals within radius $d$ of this $P_I$, and the ratio of these two corresponds to the local concentration of $P_S$; $N_S/N$ represents the global concentration of $P_S$ (please refer to the S1 File for more details).

As shown in Fig 6B, $\zeta$ gradually converges to 0 with increasing $v$, indicating that the system gradually transitions from a strongly heterogeneous system to a weakly heterogeneous system (therefore satisfying the assumption of homogeneous mixing). We can also define the critical velocity $v_c$ of $\zeta$ at which $\zeta$ reaches $0.1$($RNC_S$ reaches $0.9$). As shown in Fig 6C, the saturation velocity of $\zeta$ is comparable to that of $R_0$, and both are positively correlated with $p_{SE}$, and less correlated with $d$, initial $n_{I,0}$, $p_{EI,Day7}$, and $p_{IR}$. This is easy to understand. When $p_{SE}$ increases, the number of $P_S$ decreases while $P_I$ hardly reaches new susceptible particles, so the spatial heterogeneity is further enhanced. The spatial heterogeneity of agents other than $P_S$ is not strongly associated with $P_S$. When $d$ increases, the distribution of agents within $d$ (local distribution) is closer to the distribution of agents within the whole system (global distribution), so the spatial heterogeneity decreases. However, this effect seems not significant to $v_c$. Therefore, the trend of the two phases of $R_0$ under different velocity conditions is caused by spatial heterogeneity.

Fig 6 shows that the variation in $R_0$ with velocity shows two different phases, with the boundary of $v_c$. We further investigated the relationship between the maximum slope $k$ obtained by fitting the data before $v_c$ and the saturated reproduction number $R_s$ after $v_c$ and other propagation parameters (Fig 7). The two indexes, $k$ and $R_S$, have consistent variability, i.e., they are both positively correlated with $d$ and $p_{SE}$, negatively correlated with $p_{IR}$, and independent of the initial $n_{I,0}$ and $p_{EI,Day7}$. For $R_S$, we further find that the data can be approximately proportional to $d^2 p_{SE}/p_{IR}$. In this case, the result obtained from the CDD model is close to that of the mean field method.

Notably, the results obtained from the CDD-SEIR model show that $R_0$ is linearly related to $\log(v)$. This is not the same trend as that obtained by mean-field theory [17, 19]. One possible reason for the discrepancy obtained by the two models is that the treatment of the mean field method does not consider spatial heterogeneity, but other factors may also play a role.

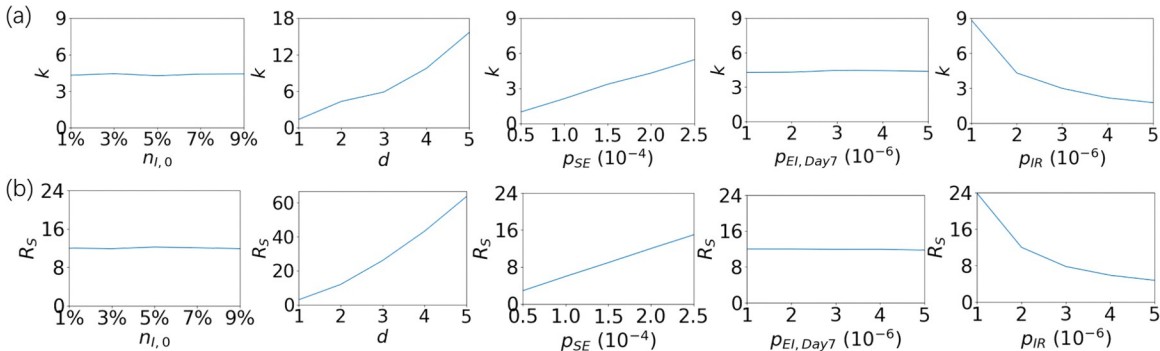

**Fig 7.** The slope $k$ (a) and the saturated reproduction number $R_S$ (b) influenced by five parameters ($n_{I,0}$, $d$, $p_{SE}$, $p_{EI,Day7}$, and $p_{IR}$) in the system.

### Effect of the CDD model on $R_0$

As shown in **Fig 6**, when $v < v_c$, the basic reproduction number $R_0$ is nearly independent of the virulence $p_{SE}$. In the work of the mean field models [19], $R_0 = (1 - e^{-\omega/\tau_T})v\sigma_0\rho\tau_I$, where $\omega$ is the exposure time, $\tau_T$ is the characteristic time of infection, $1 - e^{-\omega/\tau_T}$ represents the probability of infection in each contact, $\sigma_0$ is the scattering cross section of the agents, $\rho$ is the density, and $\tau_I$ is the transition time from infectious to recovered.

When the mobility of the agents is low, $\omega$ is much larger than $\tau_T$, and therefore, infection must occur in each contact, resulting in $R_0$ that is independent of viral virulence. From this perspective, the results obtained by the CDD-SEIR model are consistent with those obtained by the mean-field model. However, there are naturally obvious differences between the two. Spatial heterogeneity caused by the contact model of the CDD model plays an important role in the infection.

As shown in **Fig 4**, the CDD model prefers short-term contact at low velocity, while the proportion of a long contact time is low, indicating that even though the CDD-SEIR model and the mean-field model obtain qualitatively consistent trends, they are fundamentally different. The reason for this lies in the fact that differences in individual motion are considered in the CDD model so that spatial heterogeneity can be described, while the mean field treatment ignores such differences. To further illustrate the difference between these two models, we selected the cases of a relatively low velocity ($v = 0.001$ m/s, where $v < v_c$) and a medium velocity case ($v = 0.02$ m/s, where $v > v_c$) and calculated the total time $t_{SE}$ required for the $P_S$ agents to contact $P_I$ agents to become infected. The number of contacts between $P_S$ and $P_I$ before infection $T_{SI}$ and the mean and median of contact time per contact between $P_S$ and $P_I$ before infection $\omega_{SI}$ (please refer to the **S1 File** for more details) varied with the probability of infection $p_{SE}$ (**Fig 8**).

At low velocities ($v = 0.001$ m/s), the $P_S$ agent only has 2–3 contacts with $P_I$ agents before becoming infected, and the duration of each contact $\omega_{SI}$ tends to be relatively long. This

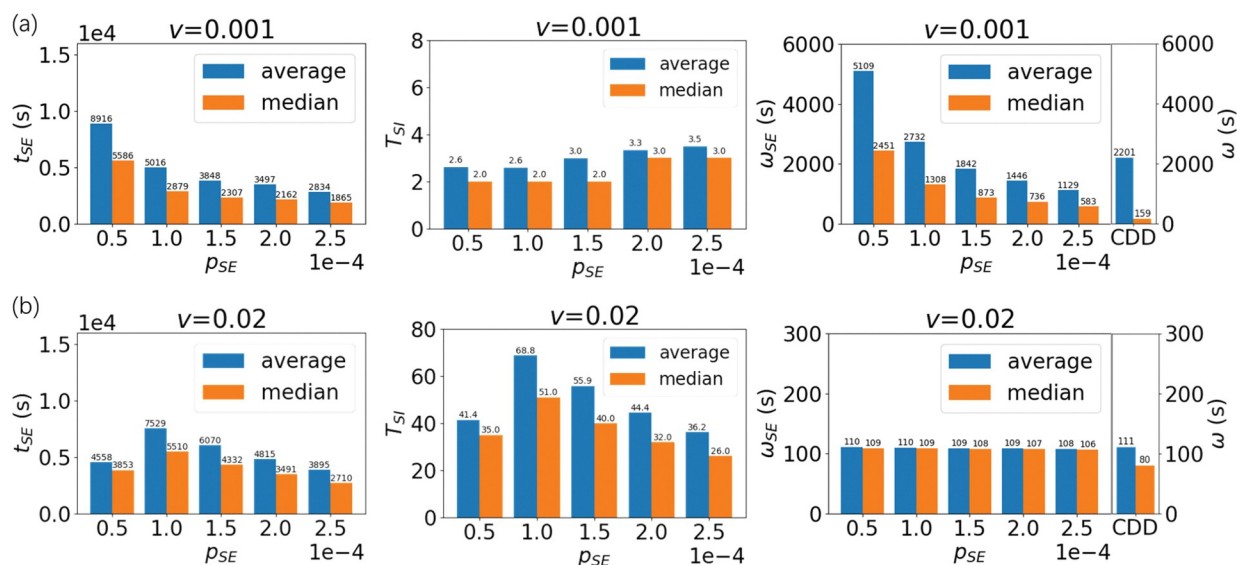

**Fig 8. Contact time required for infection $t_{SE}$ at low velocity (a, $v = 0.001$ m/s) and medium velocity (b, $v = 0.02$ m/s), number of contacts between $P_S$ and $P_I$ agents before infection $T_{SI}$, the mean (blue) and median (orange) of the average time per contact between $P_S$ and $P_I$ agents before infection $\omega_{SI}$ with the probability of infection $p_{SE}$ compared with the contact time $\omega$ calculated by the CDD model.** At a low velocity ($v = 0.001$ m/s), infection tends to occur through a small number of long-term contacts.

observation indicates that at low velocities, long-term contact is essential for the infection, and short-term contact is usually not infectious. This can be understood by the longer mean free time $\tau$ at low velocities (**Fig 4C**). From Eq 3, we can obtain $\tau \approx 0.22$ days when $v = 0.001$ m/s. In this case, even if all the surroundings are $P_I$ agents, it would take several days for $P_S$ agents to accumulate enough short exposures to become infected. However, in the real situation, the disease state of the infected person cannot be maintained indefinitely, and the whole system cannot maintain the state of many infected persons for a long time, so the contact time $t_{SE}$ obtained is often less than the ideal value of $1/p_{SE}$ (**Fig 8A**), and there may even be a simultaneous decrease in $p_{SE}$ and $t_{SE}$. Therefore, infections caused by multiple short-term contacts that take a long time to occur at low velocity are in fact very difficult to occur. This means that only a small number of prolonged exposures under the CDD model can achieve the conditions for infection occurrence in the mean field model. The long-term contact for transmission may account for strong spatial heterogeneity at low velocity.

When the velocity increases ($v = 0.02$ m/s), it is known by Eq 3 that $\tau$ will decrease significantly. In this case, the particles do not have to sift through longer collisions used for infection to occur, and infection can occur through more contacts. Meanwhile, $\omega_{SI}$ is also closer to an ordinary contact (**Fig 8B**), which is close to the results obtained by the mean-field model.

Presumably, allowing short contact exposure with good protective measures can significantly reduce $R_0$. Especially when control measures are strong, it is difficult for infections to reach new susceptible particles, and the effective ability of the virus to spread will be limited.

## Comparison with other models

Various epidemiological models that consider multiple types of compartments beyond the basic SIR framework have been proposed. In this study, we compare our CDD-SEIR method to two recently reported methods. One is a random walker's approach that incorporates the memory effect into a compartment model [37], where recovered agents are assumed to stay immune for a random duration. In contrast, our work considers the scenario where the susceptible agents become carriers after being infected and combines a modified SEIR model with contact-dependent dynamics. Both methods are based on the traditional SIR model and explore the influence of other factors on space-time dynamics of epidemics. The random walker's method is efficient for characterizing epidemics in the long term, while our CDD-SEIR model introduces more dynamical details by considering patterns of contacts. As the two methods consider different impacts of other factors on disease transmission, direct comparison of results is challenging. Future studies can further incorporate an immunity impact on transmission in our model.

The SEIRS model considers the effects of birth, death, and incubation period on infectious disease dynamics [38]. Both this model and our work take into account the latency effect, although in different ways. However, the SEIRS model integrates multiple influences, such as the length of the latent period, the various routes of transmission, and age-specific transmission. Additionally, the introduction of repeated infection in the SEIRS and random walker's models enables the epidemic equilibrium, allowing for longer timescale characterization of the disease spreading. While our CDD-SEIR model can incorporate these factors by adding agent types to simulate more realistic scenarios, we primarily focused on the impact of contact due to motion on epidemic transmission. Although some details of the epidemic transmission process are excluded, our CDD-SEIR model captures the fundamental information and physical laws.

## Conclusion

In this work, our proposed CDD-SEIR model, which combines the real-scale random motion of individuals with the traditional compartment model, discloses that epidemic transmission

depends on spatial heterogeneity and contact patterns. The CDD model has a larger proportion of short-term contacts as the velocity of agents decreases. However, we demonstrate that the contact pattern of the CDD model with short contact times at low velocity makes the virus more likely to spread through a small number of long contacts, which causes strong spatial heterogeneity and low $R_0$. Due to the presence of spatial heterogeneity, $R_0$ will be proportional to the logarithm of the group mobility in the strongly heterogeneous range and saturated in the weakly heterogeneous range, and $R_0$ is approximately independent of the virulence of the virus when the mobility of the group is low. This demonstrates that in the case of a high reproduction number, the vaccination policy and reduced contact time can still significantly inhibit the virus transmission capacity.

In the selection of parameters, we tried to make the individual parameters more adapted to the real scale, which allowed us to obtain a more accurate description of the distribution of exposure times, thus helping to explain how the movement model and spatial heterogeneity can influence the spread of epidemics from a microscopic perspective. This is something that has been relatively lacking in previous work. Such data also fully illustrate the existence of heterogeneity in the CDD-SEIR model and the difference with the mean-field model.

Overall, we used a real-scale contact-dependent motion model to reveal the effects of spatial heterogeneity and infection patterns on disease transmission from a microscopic perspective. The insight obtained from this work provides us with a new understanding of how movement patterns affect the ability of viruses to spread and how to prevent them more effectively. We hope to improve this CDD-SEIR model in the future to make the movement of individuals more realistic and thus be able to simulate some of the transmission that is difficult to characterize by numerical or network models.

## Supporting information

**S1 File. Computational details and additional data analysis.**
(DOCX)

## Author Contributions

**Conceptualization:** Hao Dong, Wei Wang.

**Data curation:** Youyuan Zhu.

**Formal analysis:** Youyuan Zhu, Ruizhe Shen, Hao Dong.

**Funding acquisition:** Hao Dong, Wei Wang.

**Investigation:** Youyuan Zhu, Ruizhe Shen, Hao Dong.

**Methodology:** Youyuan Zhu, Hao Dong.

**Project administration:** Hao Dong.

**Resources:** Hao Dong.

**Software:** Hao Dong.

**Supervision:** Hao Dong, Wei Wang.

**Validation:** Hao Dong.

**Visualization:** Hao Dong.

**Writing – original draft:** Youyuan Zhu, Hao Dong, Wei Wang.

**Writing – review & editing:** Youyuan Zhu, Hao Dong, Wei Wang.

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
