## [Decision Letter · Decision Letter 0]

21 Feb 2023

PONE-D-23-00896Spatial heterogeneity and infection patterns on epidemic transmission disclosed by a combined contact-dependent dynamics and compartmental modelPLOS ONE

Dear Dr. Dong,

Thank you for submitting your manuscript to PLOS ONE. After careful consideration, we feel that it has merit but does not fully meet PLOS ONE’s publication criteria as it currently stands. Therefore, we invite you to submit a revised version of the manuscript that addresses the points raised during the review process.

We look forward to receiving your revised manuscript.

Kind regards,

Junyuan Yang

Academic Editor

PLOS ONE

Journal Requirements:

" ext-link-type="uri" xlink:type="simple">https://journals.plos.org/plosone/s/file?id=ba62/PLOSOne_formatting_sample_title_authors_affiliations.pdf"

"This work was supported by the National Natural Science Foundation of China (Grant Nos. 12090052, 21833002), the National Key RD Program of China (2021ZD201302), and the “Fundamental Research Funds for the Central Universities” (021514380018). Parts of the calculations were performed using computational resources on an IBM Blade cluster system from the High-Performance Computing Center (HPCC) of Nanjing University."

Reviewers' comments:

Reviewer's Responses to Questions

**Comments to the Author**

1. Is the manuscript technically sound, and do the data support the conclusions?

Reviewer #1: Yes

Reviewer #2: Yes

2. Has the statistical analysis been performed appropriately and rigorously? 

Reviewer #1: N/A

Reviewer #2: Yes

3. Have the authors made all data underlying the findings in their manuscript fully available?

Reviewer #1: Yes

Reviewer #2: No

4. Is the manuscript presented in an intelligible fashion and written in standard English?

Reviewer #1: Yes

Reviewer #2: Yes

5. Review Comments to the Author

Reviewer #1: In this paper, the authors proposed a modified SEIR-CDD model, where SEIR model to describe infection between individuals and CDD model designed to characterize the stochastic motion of individuals within a community. The exposure time $\\omega$ and

mean free time $\\tau$ are discussed. Meanwhile, the epidemic transmission characteristics described by the SEIR-CDD model, the calculated reproduction number $R$ and spatial heterogeneity $\\zeta$ are obtained. And the effects of motion parameters (such as speed) on these characteristics are obtained. The bipartite variation of $R$ with the strength of spatial heterogeneity is discussed. The research content of the paper is interesting and somewhat innovative.\\\\

Reviewer #2: The paper investigates a SEIR compartemental model of epidemic dynamics. Basically it considers a population of random walkers ('agents') in a box where a transmission of the disease takes place in case of contact of an infected and a susceptible agent. The simulation is performed purely numerically, however, not sufficient details are provided to repeat these simulations. Some more information on the numerics should be included.

It seems to me that the `reproduction number R' the authors work with corresponds to the ``effective reproduction number ' Re' (number of new infections generated by one infectious agent, whereas R0 indicates this number when one infectious walker is present in a purely susceptible population). The authors should clarify this issue ad give a formal explicit definition of their `R and better explain its precise meaning'.

The paper is actually well written and the results are rather interesting. It would be nice to see a short stability analysis under which condition an epidemics is spreading (such as R-01 in standard models). What are the pertinent parameters that makes the heathy initial state (one infectious agent in a susceptible population) unstable (such as R0 1)?

In a revision the authors should also discuss possible connections to existing SEIR and other compartment models, such as a random walker's approach PhysRevE.105.024205 and existing SEIR models https://doi.org/10.1038/s41592-020-0856-2 )

6. PLOS authors have the option to publish the peer review history of their article (what does this mean?). If published, this will include your full peer review and any attached files.

Reviewer #1: No

Reviewer #2: No

---

## [Author Response · Author response to Decision Letter 0]

19 Apr 2023

Please see the attached "Response_230406" file for our point-to-point response to the comments from the two reviewers. Thank you!

---

## [Decision Letter · Decision Letter 1]

19 May 2023

Spatial heterogeneity and infection patterns on epidemic transmission disclosed by a combined contact-dependent dynamics and compartmental model

PONE-D-23-00896R1

Dear Dr. Dong,

We’re pleased to inform you that your manuscript has been judged scientifically suitable for publication and will be formally accepted for publication once it meets all outstanding technical requirements.

Kind regards,

Junyuan Yang

Academic Editor

PLOS ONE

Additional Editor Comments (optional):

Reviewers' comments:

Reviewer's Responses to Questions

**Comments to the Author**

1. If the authors have adequately addressed your comments raised in a previous round of review and you feel that this manuscript is now acceptable for publication, you may indicate that here to bypass the “Comments to the Author” section, enter your conflict of interest statement in the “Confidential to Editor” section, and submit your "Accept" recommendation.

Reviewer #1: All comments have been addressed

Reviewer #2: All comments have been addressed

2. Is the manuscript technically sound, and do the data support the conclusions?

Reviewer #1: Yes

Reviewer #2: Yes

3. Has the statistical analysis been performed appropriately and rigorously? 

Reviewer #1: No

Reviewer #2: Yes

4. Have the authors made all data underlying the findings in their manuscript fully available?

Reviewer #1: Yes

Reviewer #2: Yes

5. Is the manuscript presented in an intelligible fashion and written in standard English?

Reviewer #1: Yes

Reviewer #2: Yes

6. Review Comments to the Author

Reviewer #1: Though some questions we asked have not been answered fully satisfactorily, most of them are solved well with effort, therefore, I recommend this paper to PLos ONE.

Reviewer #2: The authors have well adressed all issues. The revised version is suitable for publication in PLOS ONE.

7. PLOS authors have the option to publish the peer review history of their article (what does this mean?). If published, this will include your full peer review and any attached files.

Reviewer #1: No

Reviewer #2: No

---

## [Editor Report · Acceptance letter]

2 Jun 2023

PONE-D-23-00896R1 

Spatial heterogeneity and infection patterns on epidemic transmission disclosed by a combined contact-dependent dynamics and compartmental model 

Dear Dr. Dong:

I'm pleased to inform you that your manuscript has been deemed suitable for publication in PLOS ONE. Congratulations! Your manuscript is now with our production department. 

Kind regards, 

on behalf of

Dr. Junyuan Yang 

Academic Editor

PLOS ONE